# Physical and Chemical Characterization of Biomineralized Collagen with Different Microstructures

**DOI:** 10.3390/jfb13020057

**Published:** 2022-05-13

**Authors:** Tianming Du, Yumiao Niu, Youjun Liu, Haisheng Yang, Aike Qiao, Xufeng Niu

**Affiliations:** 1Beijing International Science and Technology Cooperation Base for Intelligent Physiological Measurement and Clinical Transformation, Department of Biomedical Engineering, Faculty of Environment and Life, Beijing University of Technology, Beijing 100124, China; dutianming@bjut.edu.cn (T.D.); yumiaoniu@163.com (Y.N.); lyjlma@bjut.edu.cn (Y.L.); haisheng.yang@bjut.edu.cn (H.Y.); 2Key Laboratory of Biomechanics and Mechanobiology (Beihang University), Ministry of Education, Beijing Advanced Innovation Center for Biomedical Engineering, School of Biological Science and Medical Engineering, Beihang University, Beijing 100083, China

**Keywords:** mineralized collagen, microstructure, physical characterization, chemical characterization

## Abstract

Mineralized collagen is the basic unit in hierarchically organized natural bone with different structures. Polyacrylic acid (PAA) and periodic fluid shear stress (FSS) are the most common chemical and physical means to induce intrafibrillar mineralization. In the present study, non-mineralized collagen, extrafibrillar mineralized (EM) collagen, intrafibrillar mineralized (IM) collagen, and hierarchical intrafibrillar mineralized (HIM) collagen induced by PAA and FSS were prepared, respectively. The physical and chemical properties of these mineralized collagens with different microstructures were systematically investigated afterwards. Transmission electron microscopy (TEM) and scanning electron microscopy (SEM) showed that mineralized collagen with different microstructures was prepared successfully. The pore density of the mineralized collagen scaffold is higher under the action of periodic FSS. Fourier transform infrared spectroscopy (FTIR) analysis showed the formation of the hydroxyapatite (HA) crystal. A significant improvement in the pore density, hydrophilicity, enzymatic stability, and thermal stability of the mineralized collagen indicated that the IM collagen under the action of periodic FSS was beneficial for maintaining collagen activity. HIM collagen fibers, which are prepared under the co-action of periodic FSS and sodium tripolyphosphate (TPP), may pave the way for new bone substitute material applications.

## 1. Introduction

Bone, as a mechanically adaptive organ, can adapt to a variety of external forces by adjusting its complex hierarchical structure [1]. The basic unit of the complex hierarchical structure of bone is mineralized collagen, which consists of collagen fibers and apatite [2,3]. Collagen fibers are the most resourceful proteins in animals and have good mechanical sensitivity [1,4,5]. Hydroxyapatite (HA) crystals are typical representatives of apatite, which is transformed from amorphous calcium phosphate (ACP). ACP and HA crystals have different calcium (Ca): phosphorus (P) ratio ratios and crystallinity [6]. Therefore, it is an effective mechanical factor due to its higher stiffness and crystallinity [7,8,9,10]. HA can effectively promote the growth of osteoblast with high crystallinity. In the process of mineralization, collagen can induce the nucleation and crystallization of ACP, and then the ACP grow along the long axis of the aligned collagen fibers, eventually forming mineralized collagen with different microstructures [11,12,13,14,15]. Mineralized collagen is a response for the second level for the highly anisotropic hierarchical structure of bone. The highly anisotropic hierarchical structure is critical to the good mechanical and biological activity properties of bone [16].

Currently, mineralized collagen mainly includes extrafibrillar mineralized (EM) collagen and intrafibrillar mineralized (IM) collagen. Since the structure of the IM collagen is closer to the natural mineralized collagen structure, the study of the IM collagen has gained more attention. The IM collagen is divided into the normal IM collagen and the hierarchical intrafibrillar mineralized (HIM) collagen. According to the polymer-induced liquid phase precursor mineralization process, the polyanionic compounds, such as polyacrylic acid (PAA), can stabilize the ACP precursors as sequestration analogues to induce normal IM collagen [17,18]. Meanwhile, polyphosphate compounds, such as sodium tripolyphosphate (TPP), can regulate HIM collagen as template analogues with the co-presence of PAA [19,20,21,22]. However, the crystal transformation is slower and the mineralized collagen fibers are arranged poorly under the action of PAA [8,23,24]. Since the crystal size and crystallinity will affect the mechanical and biological activity properties of the materials, in order to prepare a better bone substitute material with the arrangement microstructure and the high crystal conversion degree, it is necessary to prepare an arranged HIM collagen under the action of external force.

Among the external force, the periodic fluid shear stress (FSS) is regarded as the most important mechanical stimulation mode for the bone matrix [25,26]. Correspondingly, the collagen mineralization can be guided by the external mechanical environment [27,28,29]. According to our previous study, the periodic FSS can replace the PAA to induce a highly arranged IM collagen and a highly arranged HIM collagen with the co-presence of TPP [30,31]. HIM collagen could better promote the biological responses [32]. Nevertheless, in the bone, these mineralized collagens with different microstructures may exist at the same time and result in the formation of the complex structure of the bone [33]. The physical and chemical properties of the materials are the key to the biocompatibility. The basic physical and chemical properties of these mineralized collagens with different microstructures have not been systematically analyzed.

Hence, we prepared mineralized collagen fibers with different microstructures under the action of PAA, periodic FSS, and periodic FSS-TPP, respectively. The morphology of the mineralized collagen was studied by scanning electron microscopy (SEM) and transmission electron microscopy (TEM). The molecular structure and the crystal structure of the mineralized collagen fibers was investigated by Fourier transform infrared spectroscopy (FTIR). The hydrophilicity, the stability, and the salinity of the mineralized collagen fibers were examined by analyzing the contact angles, the enzymatic times, and the thermogravimetry (TG) [34]. Our aim was to analyze the physical and chemical properties of the mineralized collagen with different mineralized microstructures, provide basic data support for the further exploration of the osteogenesis properties of those mineralized collagen fibers with different microstructures, and finally prepare advanced bone substitute materials.

## 2. Materials and Methods

### 2.1. Preparation of Mineralized Collagen with Different Microstructures

The collagen source solution was obtained from BD Biocoat (No. 354236, Corning, New York, NY, USA). The collagen was rat tail collagen (type I). The concentration of collagen was around 3–5 mg/mL. The collagen source solution was mixed with 0.1 M CaCl_2_ and 0.1 M (NH_4_)_2_HPO_4_ according to the molar ratio of Ca: P in HA. In the mineralization system, the molar ratio of calcium-to-phosphorus was kept to 1.67. The pH of the mineralization system was adjusted to 7.4 by 1 M NaOH and 0.1 M ammonium hydroxide. Then, the collagen concentration was adjusted to 1 mg/mL by distilled water, and Ca concentration was regulated to 5 mM by distilled water in the mineralization reaction system. In addition, 1 mg/mL PAA was added into the mineralization system, and its final concentration was 0 and 30 μg/mL for the preparation of EM collagen and normal IM collagen, respectively.

Meanwhile, the mineralization experiments were repeated with the same collagen and Ca concentration but adding the periodic FSS adjustment step during the mineralization process. A cone-and-plate viscometer (Brookfield R/S-CPS+ rheometer, Rheo3000, Brookfield, CT, USA) was used to provide 1.0 Pa of periodic FSS (every period is two hours, which involves work for 1 h and rest for 1 h) for the arranged IM collagen.

In addition, the collagen phosphorylated under the action of 3% TPP (Na_5_P_3_O_10_, No. T5508, Sigma-Aldrich, Saint Louis, MO, USA) for 5 h was mineralized using the same procedure to prepare the arranged HIM collagen under the co-action of periodic FSS [35].

All the mineralization reactions were kept at room temperature (RT) for 24 h. After being mineralized for 24 h, these samples were washed three times and excess water was removed by centrifugation. The samples were centrifuged at 5000 rpm for 10 min (Universal 320R, Hettich, Vlotho, Germany) and then, further lyophilized (Christ Alpha 1-4 LD, Christ, Osterode am Harz, Germany) for the following analysis.

### 2.2. Transmission Electron Microscopy (TEM) Observations

The internal morphologies of the mineralized collagen with different microstructures were analyzed by TEM (JEM-2100F, JEOL, Tokyo, Japan). The lyophilized mineralized collagen samples with different microstructures were embedded in spurr resin at 65 °C for 24 h without staining. The resin-embedded samples were cut into thin slices with a thickness of less than 100 nm by an ultraslicer (Leica UC7, speed 2.0 mm/s, Wetzlar, Germany). Ultrathin sections were transferred to copper mesh. These ultrathin samples were observed by TEM, and an analysis of the crystallization of the mineralized collagen was carried out by a selected area electron diffraction (SAED) at 200 kV.

### 2.3. Scanning Electron Microscopy (SEM) Observations

SEM (LEO1530VP, Munich, Germany) was used to observe the morphology of the mineralized collagen with different microstructures. All the lyophilized mineralized collagen samples were cut into small patches (5 × 5 mm) and fixed on the special sample stage for SEM with the conductive tapes. Then, the sample stage with the mineralized specimens was sputter-coated with a layer of gold using an ion sputter coater (SBC-12, Beijing Zhongke Instrument Co., Ltd., Beijing, China) at 0.1 Torr, 15 mA for 90 s total. Finally, the gold-sprayed sample was placed in the sample chamber of the SEM and observed under the condition of an accelerating voltage of 15 kV.

### 2.4. Fourier Transform Infrared Spectra (FTIR) Measurements

The molecular structure and the crystallinity of the mineralized collagen were analyzed by FTIR (NICOLET380, Boston, MA, USA). The FTIR measurements were performed using the potassium bromide pellet pressing method. In the tablets, the mass ratio of mineralized collagen and potassium bromide was controlled between 1:150 and 1:250. All spectra were recorded in the range of 400~4000 cm^−1^ at a resolution of 4 cm^−1^ intervals, and spectra plots represented 32 scans [36].

### 2.5. Hydrophilicity of Mineralized Collagen

The hydrophilicity of the mineralized collagen sponges can be analyzed by measuring the contact angle of the lyophilized collagen sponge surface using a contact-angle-measuring instrument (JC2000DM, Beijing Zhongyi Kexin Technology Co., Ltd., Beijing, China) [37]. The sessile drop contact angle measurements are carried out by 1 μL distilled water drops, gently delivered from a capillary tip onto the mineralized collagen surfaces. Due to the better hydrophilicity of the collagen material, we analyzed the dynamic contact angle of the different materials by taking photographs every 0.5 s for 30 s. The dynamic contact angles of the different sponges were measured three times.

### 2.6. Enzyme Measurements

A certain amount (1 mg) of the lyophilized mineralized collagen was placed in a 357 units/mL collagenase solution. The collagenase solution was prepared with Tris-HCl buffer at pH 7.4, 0.05 M. In order to simulate the enzymatic hydrolysis process in vivo, these samples with collagenase solution were dissolved in a water bath at 37 °C. The time of the dissolution of these samples can be seen as the time of enzymolysis.

### 2.7. Thermogravimetry (TG) Measurements

Weight loss can be used to analyze the mineralization degree and structural stability. The weight losses of the mineralized collagen with different microstructures (3~5 mg) were measured using a TG analyzer (LABSYS evo, Setsys, Caluire, France). According to the degradation temperature of the collagen and HA, the conditions of the TG test are as follows: the heating temperature range is from 50 °C to 800 °C, the heating rate is 20 °C min^−1^, and the atmosphere is Ar.

### 2.8. Statistical Analysis

A statistical analysis was performed using one-way analysis of variance (ANOVA) on SPSS 19.0. Corresponding *p*-values of less than 0.05 were considered significant.

## 3. Results and Discussion

During the collagen mineralization, ACP nucleates, grows, and crystallizes along collagen fibrils to form different internal mineralized structures. The internal microstructures of mineralized collagen under different conditions were observed by unstained TEM. Non-mineralized collagen is transparent under the TEM (Figure 1A). As a typical polyanionic compound, PAA is used to stabilize ACP precursors and induce the formation of nano-sized ACPs. Without the action of PAA and periodic FSS, ACP was too large to permeate into collagen fibers, large-sized ACPs attached only on the surface of collagen and formed EM collagen (Figure 1B, triangles). Meanwhile, with the action of PAA, the nano-sized ACP was formed; the nano-sized ACPs were smaller than the collagen gap regains, so they was able to penetrate into the collagen fibers through the gap regains and form normal IM collagen (Figure 1C). Arranged IM collagen formed under the regulation of periodic FSS (Figure 1D), and arranged HIM collagen formed under the co-action of periodic FSS and TPP (Figure 1E, yellow lines). The selected area electron diffraction (SAED) of the mineralized collagen produced ring patterns, proving that the ACP particles have transformed from a continuous amorphous mineral phase into a crystal phase after being mineralized for 24 h (Figure 1F). The SAED of the mineralized collagen produced crystalline patterns containing specific diffusive rings corresponding to (002), (211), and (004) reflections (Figure 1F). The specific diffusive ring patterns were the typical characteristics of highly aligned HA nanoplatelets. Under the action of PAA, normal IM collagen was randomly oriented (Figure 1C, red lines). The arranged IM collagen and HIM collagen both oriented well under the action of periodic FSS (Figure 1D,E, red lines). This proves that periodic FSS is beneficial to the neat arrangement of mineralized collagen.

SEM was used to observe the micro-spatial network structure of the mineralized collagen with different microstructures (Figure 2). All mineralized collagen sponges possess a three-dimensional interconnected porous structure, which has no significant difference before and after mineralization (Figure 2A). However, the high magnifications of these mineralized collagens (Figure 2B–F) showed the differences between these mineralized collagens. Before mineralization, the surface of collagen was very smooth. After mineralization, the surface of the EM collagen was rough, although the large ACP particles and the IM collagen were smoother than the EM collagen. Among them, we found that more nanosized spherical ACP particles were observed on the surface of the normal IM collagen induced by PAA (Figure 2D, triangle). The reason is that PAA induces intrafibrillar mineralization by stabilizing the ACPs. Furthermore, periodic FSS can reduce the size of the ACPs by its own perturbation, inducing the formation of intrafibrillar mineralization [8]. More specifically, periodic FSS can arrange the collagen fibers neatly (Figure 2E,F, arrows) and induce the mineralized phosphorylated collagen with a stripe-like structure (Figure 2F, triangles), which is the oriented HIM collagen. In addition, the oriented IM and HIM collagen under the action of periodic FSS has high pore density. The results confirmed that we have made mineralized collagen with different microstructures successfully, and periodic FSS is favorable for making the oriented HIM collagen.

FTIR spectroscopy can directly reflect the conformational structure of a compound molecule through the absorption peak of the characteristic groups. Therefore, FTIR spectroscopy was applied to measure the changes of the collagen molecule structure before and after mineralization. In the FTIR spectroscopy, amide A, amide B, amide I, amide II, and amide III are the absorption peaks of the classical structure of collagen (Figure 3a). The presence of these absorption peaks of the classical structure of the collagen triple helix structure means that the collagen triple helix structure was kept well after mineralization. During the process of mineralization, the Ca and P ions first aggregated to form ACPs; then, the ACPs transformed calcium-deficient apatite and HA crystals. ACP, calcium deficient apatite, and HA crystals have different Ca:P ratio ratios and crystallinity. The transformation of ACP to HA can be differentiated by the appearance of absorption peaks of phosphate groups around 1030 cm^−1^ and the single peak around 580 cm^−1^ gradually splitting into two peaks around 600 and 560 cm^−1^(Figure 3b, dotted lines in yellow boxes). The higher the crystallinity, the sharper the peak shape. The lower the crystallinity, the smaller the half width of the peak shape. Therefore, by observing the change of the phosphate group absorption peaks of the mineralized collagen under different conditions, we found that the characteristic peaks at 562, 600, and 1029 cm^−1^ of the collagen mineralized with the action of the periodic FSS were the highest and sharpest (Figure 3b-D,b-E arrows), indicating that the periodic FSS could promote the transformation of HA crystals and improve their crystallization rate. Meanwhile, the PAA-induced normal IM collagen has the lower absorption of characteristic peaks about the formation of HA around 560, 600, and 1030 cm^−1^ than other groups (Figure 3b-C); the reason is that PAA stabilizes the size of ACP by blocking the conversion of ACP, which in turn leads to intrafibrillar mineralization.

Furthermore, collagen, as a common biological material, has good hydrophilicity. The hydrophilic properties of the mineralized collagen are related to the network structure and the available hydrophilic groups of collagen sponges. We can analyze its hydrophilic properties after mineralization, through the dynamic contact angle. According to the changes of contact angles with increasing time, we drew dynamic contact angle curves of the mineralized collagen sponges (Figure 4). From the curves, we observed that the hydrophilic properties of the collagen sponges were good whether mineralized or not (Figure 4). The best contact angle was only maintained for 4 s; after mineralization, the contact angle can reach 10 s (Figure 4). Non-mineralized collagen has the best hydrophilic properties, and the EM collagen has the worst hydrophilic properties, indicating that this property mainly correlated to the structure of the collagen. HA or ACP particles attached on the surface of collagen can cover the hydrophilic groups, which is in accordance with the SEM analysis. Meanwhile, this study showed that the hydrophilic properties after mineralization with the action of periodic FSS were slightly less numerous than in the other groups, and the contact angles was 80°, which could be due to the arrangement of the collagen. It is worth noting that after 11 s these are all superhydrophilic, and for the proposed application, the differences in the first 11s regarding the contact angle values need to be further tested. Consistent with our TEM and SEM analysis, extrafibrillar and intrafibrillar mineralization had different levels of influence on the microstructure of the collagen sponges, which means that the contact angle values are in line with the TEM and SEM data.

Collagenases can specifically hydrolyze collagen by breaking the helical structure of the collagen and the links between the molecular chains under physiological pH and temperature conditions [38]. The enzymatic time of the collagen is reduced after mineralization. EM collagen has the lowest enzymatic time (Figure 5). The reason is that on the one hand, there are many hydrogen bonds between the helical structure of the collagen, and the hydrogen bonds can maintain the structural stability of the collagen. During the process of the HA crystal growth and deposition, the hydrogen bonding between the collagen is destroyed, resulting in the decrease of the structural stability of the collagen, which can take less time to be dissolved by collagenase. On the other hand, the larger size of the ACP, the easier it is to deposit on the surface of the collagen fibers, when mineralized collagen with the same mass, the proportion, and level of contact with the surface of collagen in the EM collagen is smaller, resulting in the enzymatic time become shorter. In addition, the PAA and the FSS all can reduce the size of the ACP. Because there is little difference between the PAA-induced normal IM collagen and the periodic FSS-induced oriented IM collagen and HIM collagen, we inferred that the conversion of the ACP has an influence on the collagen structure and enzymatic stability.

Collagen, as the organic matter in the mineralized collagen, can be gradually degraded with the increase of temperature, and the weight is obviously lost. HA, as an inorganic matter in the mineralized collagen, does not degrade with the increase of temperature, and there is no significant weight loss. Therefore, the ratio of inorganic to organic in the mineralized collagen can be determined by analyzing their weight losses. With the increase of temperature, two weight loss steps can be distinguished in the TG curves of the collagen. As the temperature increases, the binding water is lost first in the range of 30–150 °C (Figure 6 green box 1). In the range of 250–600 °C, collagen chains were broken and the structure were degraded (Figure 6 yellow box 2). The degradation of the chains means the destruction of the secondary conformation structure. Most of weight losses occur in the degradation process. Due to the fact HA can only lose the weight of the binding water, the weight loss is very slight, and the weight losses of the mineralized collagen, regardless of the mineralized microstructure, is lower than that of the non-mineralized collagen (Figure 6). Compared with the different mineralized collagens, we found that the EM collagen has the least weight losses. The oriented IM and HIM collagens have greater weight loss (Figure 6D,E). However, we found that the second stage of the weightlessness of the EM collagen appears significantly earlier, indicating that the extrafibrillar mineralization has destroyed the collagen secondary conformation structure (Figure 6B, arrow in yellow box 2). The difference in weight losses between the different mineralized collagens corresponded well with the dehydration of HA. The lower the weight losses of the mineralized collagen, the higher the content of HA in the collagen fibers. The results are consistent with the enzymatic time of the different mineralized collagens. The reason for this is that when the ACP particle sizes become smaller, their specific surface area increases, and the specific surface area of the collagen fibers is limited, resulting in the reduction of the deposition of the ACPs. Furthermore, although the perturbation of the periodic FSS can promote the formation of intrafibrillar mineralization, there is a certain obstacle to the ACP entering into the collagen fibers.

## 4. Conclusions

Since the mineralized collagen is the basic bone matrix of the hierarchically organized structures of the natural bone, there are many different mineralized microstructures of the mineralized collagen in the bone. Therefore, this study prepared the mineralized collagen with different microstructures induced under different conditions, such as the PAA and the periodic FSS. Then, we systematically analyzed their physical and chemical properties. Multiple sources of evidence were provided to prove that the addition of the PAA and the periodic FSS to the collagen mineralization system can induce intrafibrillar mineralization. The periodic FSS has improved the hydrophilicity, the enzymatic stability, and the crystal conversion of the mineralized collagen. Good hydrophilicity, enzyme stability, and crystal conversion rate are the basic conditions for the cell growth and biocompatibility of biomaterials. These experiments have laid the foundation for subsequent animal experiments and cell experiment evaluations of these mineralized collagens with different microstructures as bone substitute materials. The periodic FSS and the TPP co-induced oriented HIM collagen can be better matrix materials for the design of bone substitute materials with better bone repair abilities and may pave the way for tissue engineering applications.

## Figures and Tables

**Figure 1 jfb-13-00057-f001:**
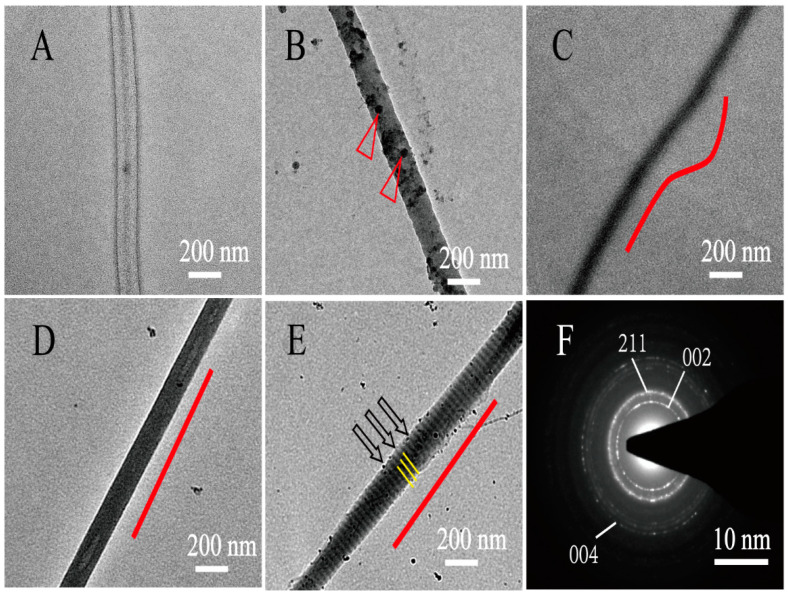
The internal morphological characteristics of different mineralized collagens observed by TEM. (**A**): The non-mineralized (NM) collagen; (**B**): the EM collagen; (**C**): the PAA-induced normal IM collagen; (**D**): the periodic FSS-induced arranged IM collagen; and (**E**): the periodic FSS and TPP-co-induced arranged HIM collagen. The NM collagen is transparent (**A**). There are some large-sized ACPs attached to the surface of the EM collagen (**B**, triangles). The IM and HIM collagen oriented under the action of the periodic FSS (**D**,**E**, red line). (**F**): The SAED of the mineralized collagen produced ring patterns. The ring patterns along (002), (211), and (004) were characteristic of highly crystalline, anisotropic HA (**F**).

**Figure 2 jfb-13-00057-f002:**
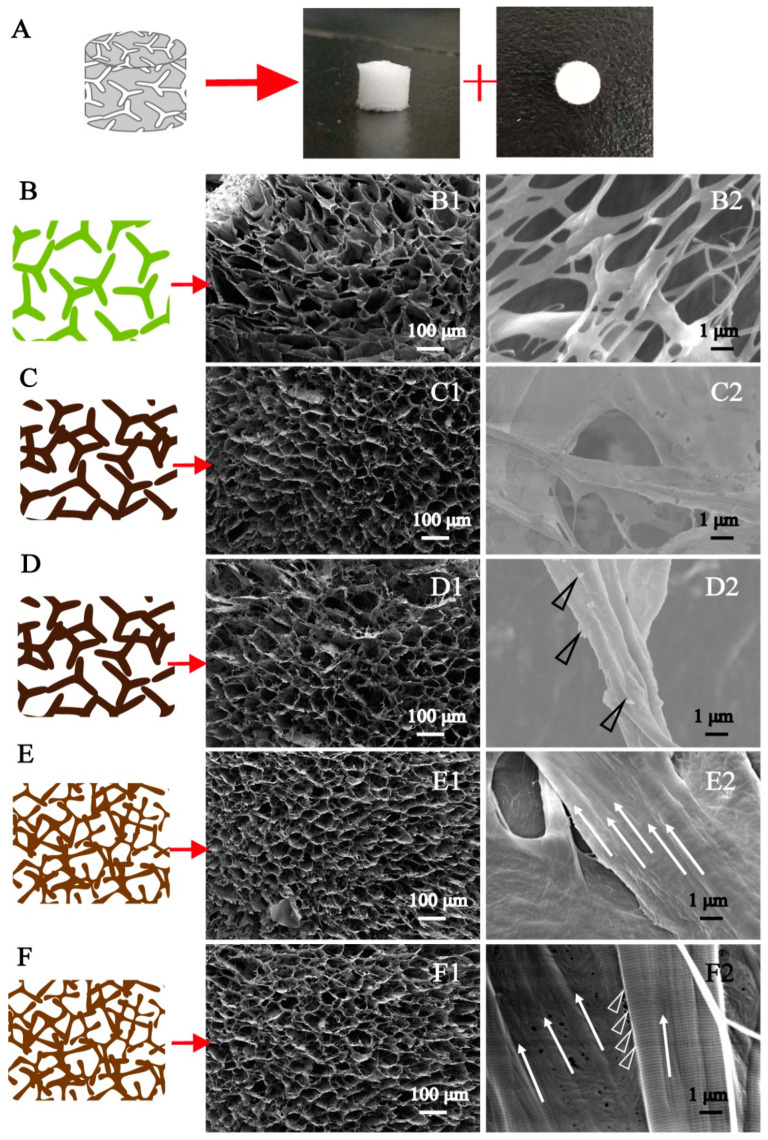
The SEM images of different mineralized collagens. (**A**): The three-dimensional structure of the collagen scaffolds; (**B**): the NM collagen; (**C**): the EM collagen; (**D**): the PAA-induced normal IM collagen; (**E**): the periodic FSS-induced oriented IM collagen; and (**F**): the periodic FSS and TPP-co-induced oriented HIM collagen. In the collagen scaffolds, the pore density has increased after mineralization, especially under the action of periodic FSS. The high magnification images of these mineralized collagens with different microstructures (**B1**–**F2**) show that the surface of the collagen is very smooth before mineralization. After mineralization, the EM collagen was rough and the IM collagen was smoother. The spherical particles were formed on the surface of the normal IM collagen induced by PAA (**D**, triangle), and collagen fibers were arranged neatly and induced by periodic FSS (**E**,**F**, arrows). The oriented HIM collagen has stripe-like structures (**F**, triangles).

**Figure 3 jfb-13-00057-f003:**
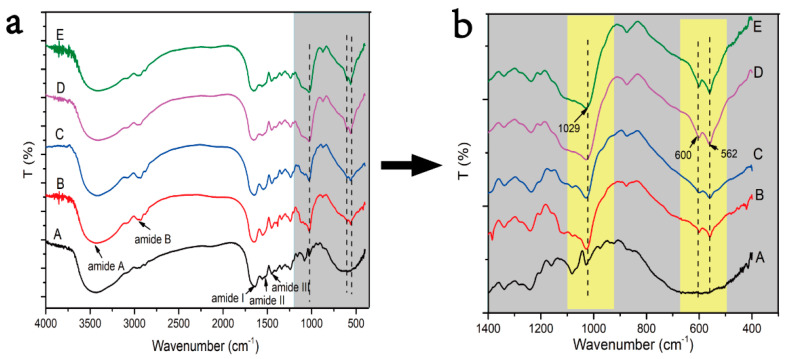
The FTIR spectra of different mineralized collagens. In (**a**,**b**), A: the NM collagen; B: the EM collagen; C: the PAA-induced normal IM collagen; D: the periodic FSS-induced oriented IM collagen; and E: the periodic FSS and TPP-co-induced oriented HIM collagen. (**b**) is an enlargement of the gray box in (**a**). The characteristic peaks of HA at 562, 600, and 1029 cm^−1^ were the highest and sharpest under the condition of periodic FSS (D and E).

**Figure 4 jfb-13-00057-f004:**
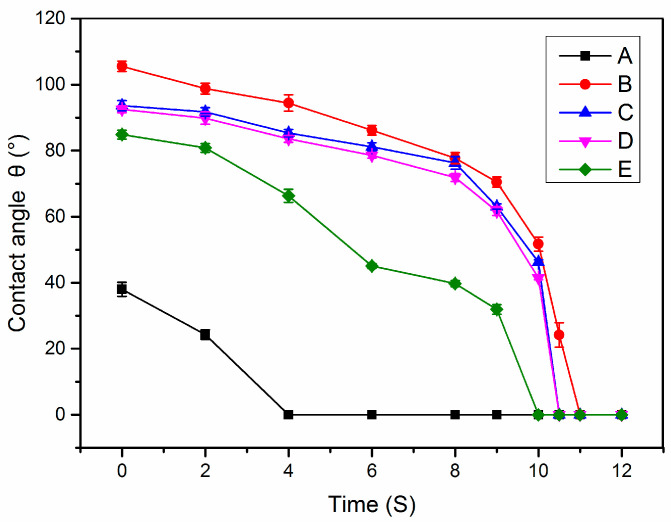
The dynamic contact angles of the different mineralized collagens. A: the NM collagen; B: the EM collagen; C: the PAA-induced normal IM collagen; D: the periodic FSS-induced oriented IM collagen; and E: the periodic FSS and TPP-co-induced oriented HIM collagen. The change of contact angles with the time was quick from 40° to 0°. The EM collagen has poorer hydrophilicity than IM collagen. The periodic FSS and TPP-co-induced oriented HIM collagen have the best hydrophilicity after mineralization, and their contact angles are 80°.

**Figure 5 jfb-13-00057-f005:**
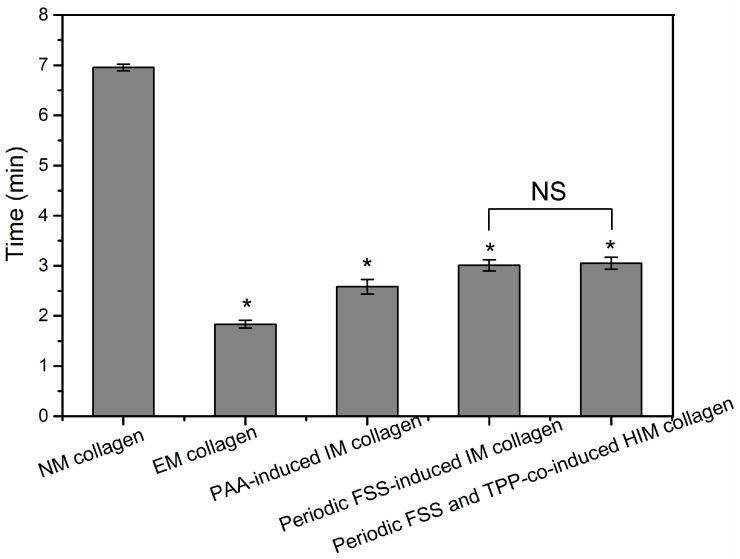
The enzymatic time of the different mineralized collagens. The NM collagen has the longest enzymatic time, and the EM collagen has the shortest enzymatic time. The enzymatic time was longer than other mineralization conditions under the action of periodic FSS. * *p* < 0.05 vs. non-mineralized collagen group, NS: no significant difference.

**Figure 6 jfb-13-00057-f006:**
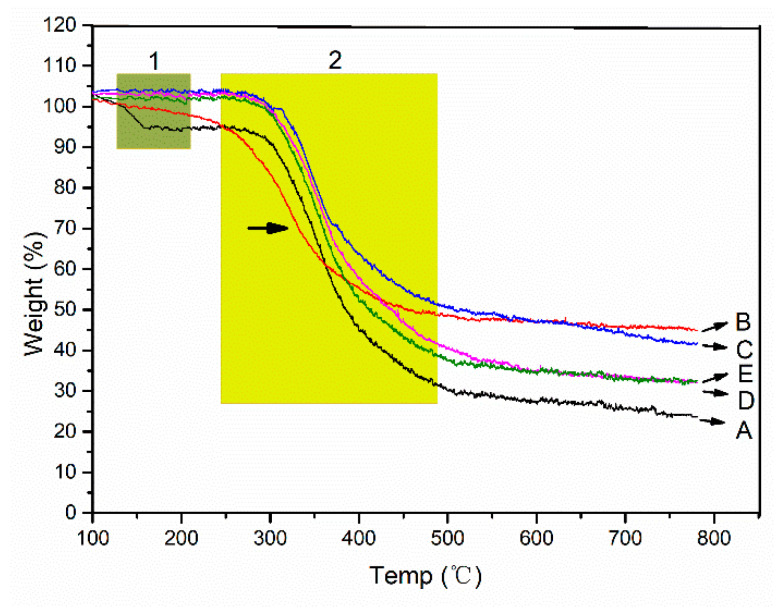
The TG curves of the different mineralized collagens. The curves from A to E were the NM collagen, the EM collagen, the PAA-induced normal IM collagen, the periodic FSS-induced oriented IM collagen, and the periodic FSS and TPP-co-induced oriented HIM collagen. The NM collagen has the largest weight losses, and the EM collagen has the smallest weight losses. The EM collagen first begins the second stage of the weight loss.

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
