# Peer review of "Physical and Chemical Characterization of Biomineralized Collagen with Different Microstructures"

_jfb, 2022, doi:10.3390/jfb13020057_

Round 1

Reviewer 1 Report

This manuscript aims to analyze the physical, morphological, structural and chemical properties of mineralized collagen with different methodologies and provide support for further exploration of the osteogenic properties of mineralized collagen fibers for bone tissue regeneration. Overall the study appears to be detailed and well performed. The characterization of these different mineralized collagen could yield some useful information for preparing of advanced bone substitute materials.

I have some comments as listed below:  

  1. In "Introduction", authors could give more clear background information about currently used various mineralization methods, including polymer-induced liquid phase precursor mineralization process such as PAA and FSS. Why these methods are used in this study could be better explained. Furthermore, it is not clear whether the aim of the study is to mimic the naturally mineralized collagen in bone or to explore various mineralization methods for bone tissue engineering.
  2. If the natural bone is highly anisotropic in hierarchical structure, what could this study contribute to further exploration of the building advanced bone substitutes.
  3. What are the different conditions lead to "Periodic FSS-induced intrafibrillar mineralized (IM) collagen" and "Periodic FSS-induced hierarchical intrafibrillar mineralized (HIM) collagen"? More detailed are needed in methodology section.
  4. For figure 1D and 1E, it is said "IM and HIM collagen oriented under the action of periodic FSS (D and E, red line)". Are they from the same mineralization condition or from different conditions?
  5. Figure 1C is blurry and I am not certain what it intends to show?
  6. More discussion is needed to explain the results and the significance of these results.

Reviewer 2 Report

Dear authors,

to my opinion your manuscript reports in the most adeguate manner your work. The abstract provides a sufficient background of the paper. 

Materials and methods are  described.

Could you please uniform the format of the characters at the lines 11.16.17.19.26.28.39.43.44.52.57.58.302 and 308.

Reviewer 3 Report

The manuscript of Du et al. reports on the biomineralization of collages with different microstructures and evaluates their different physical and chemical properties. The manuscript does fit the special issue it was submitted to, as it focus on the analysis for a possible new bone substitute material. There are however some additional points which can be improved, as mentioned below:

  1. authors should carefully check the formatting of the article, e.g. abstract, conclusion, text is not uniform.
  2. Figure 3 - FTIR data. Author could improve the figure arrangement so that the peaks and denotations belonging to HA are more visible (peak values from figure caption and text should fit those showing on the graph).
  3. Authors can also improve the naming of the samples. E.g., Fig 2 - different scaffolds B-F, Fig 2 - A-E, Fig 3 - A-E
  4. Regarding the dynamic contact angle, while in initial time or 2s of the contact angle measurements, the different morphologies have different contact angle values, after 11 s these are all superhydrophilic. Would the differences in the first 2s really matter that much for the proposed application?
  5. Are the different formation processes affecting the HA properties in view of Ca:P ratio?
  6. Are the contact angle values and other properties of these microstructures in line with other mineralized collagen data from literature?

Round 2

Reviewer 1 Report

The authors addressed most of the questions and made significant improvement overall. 

Author Response

We are so appreciated for your efforts in the revision process of our manuscript. We have checked the english of our manuscript.

Reviewer 3 Report

The authors reviesed the manuscript accordingly. However, a statement or more detailes with respect to the "the different formation processes affecting the HA properties in view of Ca:P ratio" could also be discussed into the manuscript to give a more detailed overview to the readers. For point  6, as "the contact angle values are in line with other mineralized collagen data." authors could include this statement and appropiate refernces into the reviesed manuscript.  Additionally, some minor English corrections are still needed.
